# Carbapenem-resistant enterobacteriaceae: analyzing knowledge and practice in healthcare providers

Evangeline Thibodeau[1], Shira Doron[1], Vito Iacoviello[2], Jennifer Schimmel[3] and David R. Snydman[1]

[1] Division of Geographic Medicine and Infectious Diseases, Tufts Medical Center and Tufts University School of Medicine, Boston, MA, USA
[2] Saint Elizabeth's Medical Center, Brighton, MA, USA
[3] Baystate Medical Center, Springfield, MA, USA

## ABSTRACT

**Background.** Gram negative antibiotic resistance is increasing worldwide as both carbapenem-resistant enterobacteriaceae (CREs) and *Enterobacteriaceae* producing extended spectrum ß-lactamases (ESBLs) become more common.

**Objective.** We analyzed clinicians' knowledge regarding resistant gram-negative organisms with respect to infection control practices, prescribing practices and assessment of their patients' risk for resistant infections.

**Design.** Online survey.

**Participants.** Target population included clinicians who prescribe antibiotics i.e., medical doctors and mid-level practitioners, at three Massachusetts hospitals.

**Methods.** Questionnaires were sent to 3 Tufts-affiliated teaching hospitals to assess level of knowledge and elucidate perceptions about gram-negative resistance.

**Results.** We received 434 responses from 3332 non-infectious disease clinicians (13%) surveyed at the three hospitals. 51.1% of clinicians correctly scored 50% or greater on the knowledge questions. Internal medicine clinicians had higher knowledge scores than non-internal medicine clinicians (62% vs 45%; OR = 1.67, $p = 0.02$). Clinicians within three years of training had higher scores than those with more than 10 years of training (64.3% vs 44%; OR = 2.3, $p = 0.002$). Clinicians with fewer years since training and those with higher knowledge scores were more likely to appropriately consider certain patients at risk for resistant infections ($p < 0.05$). 54.4% of clinicians were very concerned about gram-negative antibiotic resistance. 64.6% of clinicians felt comfortable de-escalating antibiotics as cultures are available.

**Conclusion.** We found overall low knowledge scores and much variability in the way clinicians assess whether certain patient populations are at risk for antibiotic resistance. Internal medicine clinicians and those with fewer years since completion of their training scored higher and more appropriately considered patients at risk for resistance. The majority of clinicians are concerned about gram-negative resistance and indicated they would de-escalate antibiotic therapy if they had susceptibility information. These results will help focus and target our teaching and awareness-raising strategies.

Corresponding author
Shira Doron,
sdoron@tuftsmedicalcenter.org

## BACKGROUND

Antibiotic resistance is increasing worldwide. While much focus has been on gram-positive organisms such as methicillin-resistant *Staphylococcus aureus* (MRSA), concern is growing regarding more extensive antimicrobial resistance in gram-negative organisms. Carbapenems, including imipenem, ertapenem, meropenem, and now doripenem, have been used increasingly over the past decade to treat infections due to *Enterobacteriaceae* producing extended spectrum ß-lactamases (ESBLs). Emergence of carbapenem-resistant *Enterobacteriaceae* (CRE) is worrisome, particularly since there are limited antibiotic options to treat such infections, many of which are associated with significant adverse events (*Nordmann, Cuzon & Naas, 2009*). Not only are drug options limited, it has been shown that patients infected with CREs suffer a 3-fold increased mortality compared to patients with infection due to a susceptible strain (*Patel et al., 2008*).

While infection control practices have been shown to decrease the spread of resistance during outbreaks (*Munoz-Price et al., 2010*), given the limited antibiotic choices to treat these infections, awareness and prevention by clinicians is imperative in preventing further spread of this epidemic. Five-hundred and three physicians were surveyed in a University in France regarding antimicrobial resistance: 98% of physicians identified antimicrobial resistance as a national problem, yet only 74% of surveyed physicians felt it affected their daily practice. Interns reported more training in antibiotic resistance than senior physicians (59% vs 34%) (*Naqvi & Pulcini, 2010*). This study focused on MRSA, a gram positive organism, rather than gram-negative resistance such as ESBLs or CREs. There is a need to better understand the general knowledge and practice of healthcare practitioners in relation to the more recent and rapidly evolving gram-negative resistance problem. This information can be used so antimicrobial stewardship teams and infection disease specialists may identify knowledge gaps and inappropriate practices to better focus their educational efforts. With improved education regarding appropriate risk assessment and prescribing practice, the further development of resistance may be slowed.

## METHODS

We conducted an online survey using the SurveyMonkey® platform to better understand the knowledge and practice of health care practitioners regarding resistant gram-negative organisms. Responses were collected from September, 2011 through January 2012. The survey was sent to three hospitals in the Boston, Massachusetts metropolitan area: Tufts Medical Center, Saint Elizabeth's Medical Center, and Baystate Medical Center. Each hospital is an academic institution affiliated with Tufts Medical School. Tufts Medical Center is a 415 bed tertiary care center located in downtown Boston. Saint Elizabeth's Medical Center is a 272 bed hospital located in a Boston suburb. Baystate Medical Center is a 716 bed facility located in western Massachusetts. During the year of the survey, in 2011, the rate of ESBL identification amongst isolates of *Escherichia coli*, *Klebsiella oxytoca*, and *Klebsiella pneumoniae* at these three hospitals ranged from 3 to 6% and there were no CREs identified. The survey was approved by all hospitals' respective International Review Boards. Consent was inferred when a subject completed the survey. The survey was sent

**Table 1  Characteristics of survey respondents (N = 419).**

| Characteristic | No. (%) |
|---|---|
| **Hospital** | |
| Tufts Medical Center | 194 (46.3) |
| Baystate Medical Center | 177 (42.2) |
| Saint Elizabeth's Medical Center | 48 (11.5) |
| **MD/DO** | 372 (88.8) |
| **Years of practice** | |
| <3 | 129 (30.8) |
| 3–10 | 106 (25.3) |
| >10 | 184 (43.9) |
| **Specialty**[*] | |
| Internal medicine | 213 (51.1) |
| Surgery | 46 (11) |
| Pediatrics | 67 (16.1) |
| Ob/Gyn | 26 (6.2) |
| Other (Psychiatry, Neurology, Radiology, Anesthesia, Radition/Oncology, Emergency medicine, PMR) | 65 (15.6) |

Notes.

[*] Note two missing values from specialty (N = 417).

electronically by email list serves, accessed through the respective employee affairs' offices. In order to increase survey response, participants were entered into a raffle to win an Amazon gift card. Data was analyzed using SPSS for frequency tables and SAS 9.2 for multivariable logistic regression. Please see Supplemental Information for the survey in its entirety.

## RESULTS

The survey was sent to 3332 clinicians at the three hospitals and 434 (13%) complete responses were received. Note 15 responses from Infectious Disease physicians were removed leaving 419 responses for analysis (194 from Tufts Medical Center, 177 from Baystate Medical Center and 48 from Saint Elizabeth's Medical Center). The survey was analyzed in the 5 following sections (in order of taking the survey): knowledge, opinion, risk perception, prescribing practices, review practices.

The characteristics of the respondents are displayed in Table 1.

### Knowledge assessment

Knowledge questions were analyzed using a composite score from the six survey knowledge questions: respondents were divided into those that scored 50% or higher on the composite score and those that did not. Overall, 51.1% of clinicians scored 50% or higher on the knowledge questions (range 0%–100%). However, 62% of internal medicine (IM) trained clinicians scored 50% or higher on their composite knowledge score compared to 45% of non-IM trained clinicians (OR = 1.67, $p = 0.02$). In addition, a significantly larger percentage of clinicians who were within 3 years since completion of their training scored

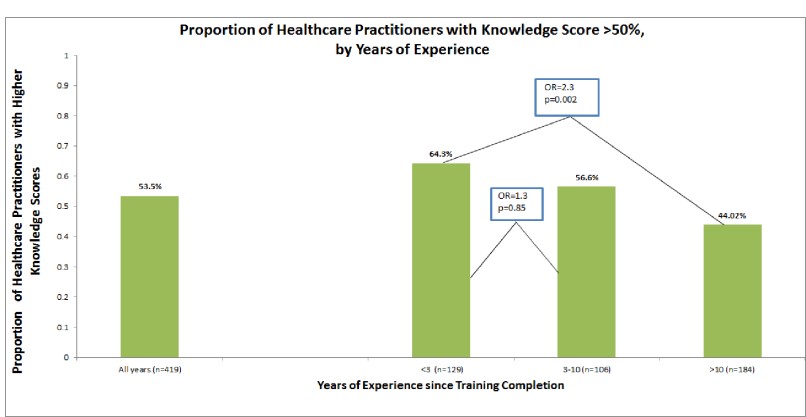

**Figure 1  Proportion of healthcare practitioners with knowledge scores >50%, by years of experience.**

50% or higher on their composite knowledge score compared to practitioners with >10 years of experience (Fig. 1, OR = 2.3, $p = 0.002$); there was no difference in higher scores between physicians within 3 years since completion compared with clinicians within 3–10 years since completion ($p = 0.85$).

## Opinion assessment

The majority of respondents were concerned about gram-negative resistance with 54.4% very concerned, and just 3.6% not knowing about gram-negative resistance prior to taking the survey. The majority of respondents (51.6%) did not agree with the following statement: "There are sufficient new antibiotics in development to treat resistant gram negative infections", however 21.6% of clinicians reported having "no idea" what their opinion was regarding the statement.

## Risk perception assessment

The majority of clinicians correctly identified patients at risk for resistant infections: dialysis patients (69.5%), patients residing in nursing homes (66.9%), and patients admitted within the past 30 days (66.2%), however overall scores were still low (Fig. 2). There was a correlation between knowledge scores and risk consideration with a higher proportion of clinicians who scored higher on the knowledge questions correctly considering patients at risk for resistance compared with clinicians who scored lower on the knowledge questions. For example, 66.8% of clinicians with higher knowledge scores correctly identified dialysis patients at risk for resistance compared with 51% of clinicians with lower knowledge scores (OR = 1.6, $p = 0.02$). This pattern was also demonstrated in clinicians recently completing training; clinicians within 3 years of completing training were more likely to correctly identify patients at risk for infections compared to clinicians more than 10 years out of training. There was no difference in risk consideration practice between clinicians within 3 years since training completion compared with those within 3–10 years since training completion, except when considering nursing home patients.

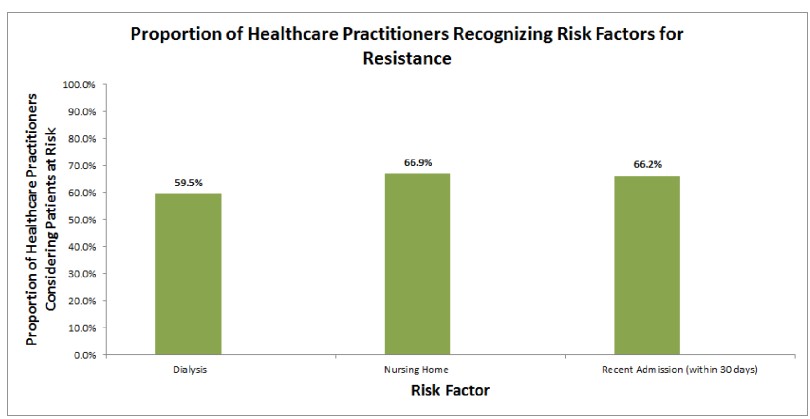

**Figure 2** Proportion of healthcare practitioners recognizing risk factors for resistance.

## Prescribing practices

When prescribing antibiotics, 81.5% of clinicians consider if their patients are at risk for resistant infections at least half the time; only 1.9% of clinicians never consider if their patients are at risk for resistant infections. The most common means of risk consideration is to review prior microbiology reports (77.5% of clinicians) followed by consideration of recent contact with healthcare environment (67%), review of prior antibiotic exposures (65%), and review of prior notes or discharge summaries (50.2%). In addition, 71.2% of clinicians call a specialist such as infectious disease or antimicrobial management teams less than half the time when prescribing antibiotics, and only 2% always call a specialist when prescribing antibiotics.

## Review practices

When prescribing antibiotics, the majority of clinicians review antimicrobial history. While 86.4% of clinicians review antimicrobial reports pertaining to the current infection at least 50% of the time, 62.4% of practitioners review antimicrobials reports that do not pertain to the current infection (old reports) at least 50% of the time. The most common reason for not reviewing the reports was when the records are not available.

When reviewing microbiological records, the majority of clinicians (57.3%) reported using the susceptibility designation ("susceptible/intermediate/resistant") to determine what antimicrobial to order, and only 34.4% of clinicians reported looking at mean inhibitory concentration (MIC) values.

In addition, 64.6% of clinicians felt comfortable de-escalating antibiotics as cultures are available.

## DISCUSSION

In the United States at least two million people are infected with resistant organisms resulting in 23,000 deaths, with most deaths occurring in healthcare settings (*Center for Disease Control and Prevention , 2013*). Given health care practitioners are at the cornerstone of antibiotic prescribing, caring for patients admitted to the hospital, and assessing their

risk for resistant infections, it is imperative to understand clinicians' baseline knowledge, prescribing practices, and risk assessment strategies. In this analysis we found that overall, health care practitioners in internal medicine were more knowledgeable about ESBL and CRE definitions and treatments compared to clinicians not employed in internal medicine. We also found that clinicians with fewer years since completion of their training had more knowledge than clinicians farther out from training.

It appears adequate concern exists regarding gram-negative resistance, however, there is still a large proportion of clinicians that are not aware about the lack of new drugs in development to treat such infections.

While the majority of clinicians consider if their patients are at risk for resistant infections prior to prescribing antibiotics, and use the appropriate means to consider patients' risk, the type of patients they consider at risk is variable. Most clinicians appropriately considered dialysis patients, nursing home patients, and patients admitted within the previous 30 days at risk for resistant infections, however, higher knowledge scores and fewer years of experience correlated with appropriate consideration of some groups of patients at risk for resistant infections. This validates our findings that not only are internal medicine trained clinicians and those with fewer years since training completion more knowledgeable about ESBL and CRE definitions, they are more appropriately considering the patients at risk for these resistant infections.

When reviewing reports, clinicians often review reports pertaining to the current infection, but do not review old reports sufficiently. In addition, the majority of clinicians do not interpret MIC values, which would not be expected from a non-infectious disease specialist. However, given the majority of clinicians do not call a specialist when prescribing antibiotics and feel comfortable de-escalating antibiotics, perhaps educational efforts should be directed toward MIC interpretation.

Limitations include the survey sampled. While previous studies have demonstrated >50% of hospitals in Massachusetts have identified CREs (*Thibodeau et al., 2012*), higher rates exist in areas such as New York City. While we would expect higher levels of awareness in such areas, given the increasing concern for resistance and rapid spread, it is important to educate early and prior to resistance rates reaching epidemic thresholds.

In summary, we found many gaps in knowledge, and a great deal of variability in opinions, and risk assessment practices, in health care practitioners in regards to resistant gram-negative infections. While it is not surprising that internal medicine employees performed better overall, with higher knowledge scores and more appropriate risk perception, it is enlightening that clinicians with fewer years since training completion fared better in many areas than clinicians with more years of experience, which likely relates to the relatively new and increasing concern for antibiotic resistance. The information from this survey will help focus and target our teaching and awareness-raising strategies through existing programs such as orientation teachings sessions, resident and attending targeted education conferences, and weekly emails.

## Funding

This study was supported by NIH Training Grant 5T32AI007329-17 and was supported in part by a research grant from the Investigator-Initiated Studies Program of Merck Sharp & Dohme Corp. The opinions expressed in this paper are those of the authors and do not necessarily represent those of Merck Sharp & Dohme Corp. The funders had no role in study design, data collection and analysis, decision to publish, or preparation of the manuscript.

## Grant Disclosures

The following grant information was disclosed by the authors:
NIH Training Grant: 5T32AI007329-17.
Investigator-Initiated Studies Program of Merck Sharp & Dohme Corp.

## Competing Interests

Shira Doron is on the speakers' bureau for Optimer, Forest and Merck, has received research funding from Merck, and has received consulting fees from Durata. Vito Iacoviello was an HIV clinical consultant for Gilead Pharmaceuticals. David R. Snydman is on the speaker's bureau for Merck, Cubist and Genentech, has received research funds from Merck, Cubist, Forest, Astra Zeneca, Replidyne, Optimer, Pfizer and Genentech, has been a consultant for CSL Behring, Genentech, Millenium, Genzyme, Boeringer Ingelheim, Massachusetts Biologic Public Health Laboratories, Merck and Microbiotix, and has provided expert testimony on behalf of Roche.

## Author Contributions

- Evangeline Thibodeau conceived and designed the experiments, performed the experiments, analyzed the data, contributed reagents/materials/analysis tools, wrote the paper, prepared figures and/or tables, reviewed drafts of the paper, study design, survey design, deploy survey, analyze data, prepare manuscript.
- Shira Doron conceived and designed the experiments, wrote the paper, prepared figures and/or tables, reviewed drafts of the paper, study design, manuscript preparation.
- Vito Iacoviello and Jennifer Schimmel conceived and designed the experiments, performed the experiments, wrote the paper, reviewed drafts of the paper, study design, facilitate deployment of survey.
- David R. Snydman reviewed drafts of the paper, study design.

## Human Ethics

The following information was supplied relating to ethical approvals (i.e., approving body and any reference numbers):

Tufts Medical Center, Saint Elizabeth's Medical Center, and Baystate Medical Center IRBs approved this study. Approval letters were provided by each IRB.

## Supplemental Information

Supplemental information for this article can be found online at http://dx.doi.org/10.7717/peerj.405.

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
