# Peer review of "Carbapenem-resistant enterobacteriaceae: analyzing knowledge and practice in healthcare providers"

_PeerJ, doi:10.7717/peerj.405_

## Round 0.1 · original submission · Major Revisions

· Academic Editor

Major Revisions

As you my see, we got somewhat divergent opinions (which also explains why the paper was a bit delayed). I strongly advise you to take ALL comments into consideration and see how you can mody the paper accordingly.

Reviewer 1 ·

Basic reporting

The main problem of this work is the presentation of the results.
The Table and Figures provided are of low quality and do not well summarize all of the results obtained. In contrast, the authors wrote a very long results section where the reader is going to get lost and then also bored.
I strongly suggest to present very readable tables/figures where the results are clearly presented. The results text section should only summarize and highlight the most important findings.

Experimental design

The methodology of the study is not well explained.
Many essential data are missed. For instance, 1) When the survey was performed? 2) What are the characteristics of the 3 hospitals? 3) How many answers were from Hospi A, B, or C?
It would be great see the template of the questions presented to the participants (supplemental data?).

Validity of the findings

Not assessable because of the insufficient information provided in the methods section.

Additional comments

Along with the critiques listed in the other sections, I would suggest modifying the introduction a bit. In particular, I would not focus on KPC and NDM but more on the aspects that this work wants to analyze.

Reviewer 2 ·

Basic reporting

The article is well written and the methodology is relatively basic. It is a nice assessment of gram negative resistance awareness amongst hospital clinicians.

Experimental design

the design is basic survey, were any commercial electronic survey tools used, e.g., "Survey Monkey"? if so, it should be stated. It would be nice to include in an online supplement the actual survey that was sent out to the clinicians. Also given that in the metro Boston area the rates of KPC and NDM-1 are low in comparison to other metropolitan areas on the east coast (i.e., NYC) it would be interesting to see the survey response from hospitals where these types of resistance issues may be more prevalent.

Validity of the findings

The findings are interesting, however the response rate was low but within the 10-20% typical seen with email surveys. Were there any incentives offered to the clinicians to complete the survey? it may have helped increased the number of respondents. if not it should be stated. It was a good assessment of the awareness of gram negative resistance issues by non-ID clinicians, it would be nice to include some examples of which types of educational activities are effective at increasing awareness about gram negative resistance issues amongst non ID clinicians. As well as which ones the authors are planning on utilizing.

Additional comments

If you have access to the antibiograms from the three institutions that were surveyed it might be nice to include the gram negtative resistance rates within each institution. if the rates are low it could explain some of the low knowledge responses, however since the actual survey was not included in this review, i am not sure if antibiogram awareness was addressed in the survey.

---

## Round 0.2 · accepted · Accept

· Academic Editor

Accept

I apologize for the delay in finalizing our decision due to the necessity to gather all opinions.

Reviewer 2 ·

Basic reporting

The article is well written and address an issue that all clinicians should be cognizant about.

Experimental design

Methods are appropriate for a survey study and their response rate (13%) was within the 10-20% response rates typically seen with survey studies and it was approved by the IRB.

Validity of the findings

The findings are not that surprising, that older clinicians have less awareness of antibiotic resistance, as this issue has become more pressing in the last 10 years due increased prevalence of resistant pathogens and fewer antimicrobials being developed to treat these infections. It would be nice to know, how many of the institutions surveyed have an active antimicrobial stewardship program and if yes, what percentage of those surveyed knew about the program.

Additional comments

In addition to the educational and quality improvement programs being proposed to increase clinician awareness of antibiotic resistance, the development of clinical pathways and guidelines are an important component.